# The Honey Bee Colony’s Criterion for Candidate Selection: “Ongoing” or “One-Shot”?

**DOI:** 10.3390/ani14111535

**Published:** 2024-05-22

**Authors:** Luxia Pan, Shiqing Zhong, Tianyu Xu, Weixuan Chen, Zhijiang Zeng

**Affiliations:** 1Honeybee Research Institute, Jiangxi Agricultural University, Nanchang 330045, China; luxia_pan@163.com (L.P.); shiqing_zhong@163.com (S.Z.); xty0111@126.com (T.X.); a1422878379@gmail.com (W.C.); 2Jiangxi Province Key Laboratory of Honeybee Biology and Beekeeping, Nanchang 330045, China; 3Laboratory Animal Science and Technology Center, Jiangxi University of Chinese Medicine, Nanchang 330004, China

**Keywords:** queenless colony, queen candidates, larval age, quality control

## Abstract

**Simple Summary:**

The honey bee is a typical social insect, and a colony is composed of a queen, workers, and drones. The queen is responsible for reproduction and is essential for the development of the colony. When the queen dies or is injured, workers raise closely related larvae to develop into a queen. Without nepotism, how do colonies choose queen candidates during emergency queen rearing? Are there criteria for queen candidates? Is the colony’s criterion for candidate selection “ongoing” or “one-shot”? Our investigation focused on the emergency queen-rearing process in a natural colony (without nepotism), where we observed and documented the selection of queen candidates from different stages (the larval stage, capping stage, and emerging stage). We examined the physiological indicators and the expression of ovarian development-related genes (*vg*, *hex110*, and *Jh*), and found that the colony would eliminate queens of low reproductive quality and prefer queens with higher reproductive ability. While the exact mechanism by which workers assess queen candidates’ quality is unknown, the result of this layered filtering of investments in queen candidates may be to maximize colony growth. In the absence of nepotism, the selection of queen candidates is not final but is gradually optimized.

**Abstract:**

In the honey bee, the queen’s death severely threatens the survival of the colony. In an emergency, new queens are reared from young worker larvae, where nepotism is thought to influence the choice of queen candidates by the workers. This article simulates the emergency queen-rearing process in a colony under natural conditions and records the results of colony selection (without nepotism). In queenless colonies, worker larvae aged three days or younger were preferred for queen rearing, and 1-day-old larvae were the first to be selected for the queen-cell cups. In the capping stage, the number of capped queen cells selected from the 1-day-old larvae was much higher than the 3-day-old larvae. On the first day, the number of emerging queens reared from 1-day-old larvae was significantly higher than the queens reared from 2-day-old and 3-day-old larvae. However, there was no significant difference in the birth weights of queens reared from 1-day-old, 2-day-old, or 3-day-old larvae. When the newly emerged queens were introduced into the original queenless colony, 1-day-old larval queens triggered more worker followers than 2-day-old larval queens. The expression of ovarian development-related genes (*vg*, *hex110*, and *Jh*) was higher in queens reared from 1-day-old larvae than those reared from 2-day-old and 3-day-old larvae, indicating that the quality of the queens reared from 1-day-old larvae is superior. This study shows that in the absence of nepotism, the colony selection of queen candidates at the larval stage, capping stage, and emerging stage is not final, but is gradually optimized to maximize colony development through a “quality control” process.

## 1. Introduction

The honey bee queen’s death severely threatens the survival of the colony. To solve this potential problem, honey bees have evolved the ability to breed replacement queens, where in an emergency, new queens are reared from young worker larvae. This process was first demonstrated by Schirack in the late 17th century [1,2], but despite these early works, the process of emergency queen production is not well studied, especially the colony’s criteria for selecting queen candidates.

Social insects use kinship to promote cooperation [3], but in honey bees (*Apis mellifera* L.), queen breeding can result in conflict due to polygamy [4,5]. In the same subfamily, the average relatedness among worker members is 0.75 (super-sister) [6], the average relatedness between workers and their reproductive offspring (drones) is only 0.5, while the average relatedness between workers in different subfamilies is only 0.25 (half-sisters) [7,8]. This view has led to studies on nepotism as a criterion for workers’ selection of larvae for the rearing of new queens [9]. When the colony is to breed a queen, the workers are faced with choosing between different nepotism indices (r = 0.75, r = 0.5, r = 0.25) for egg and larval breeding [10,11,12]. It is demonstrated that each worker is inclined to cultivate more close sister queens from its paternal line (r = 0.75) through nepotism, rather than raising less related half-sister queens from another paternal line (r = 0.25) [13]. This phenomenon often occurs when the colony’s queen dies/disappears or stops laying eggs, resulting in an emergency queen-rearing situation.

In the absence of nepotism, how do colonies choose queen candidates during emergency queen rearing? Are there criteria for queen candidates and is the colony’s criterion for candidate selection “ongoing” or “one-shot”? To investigate these questions, we chose to study queen rearing in the absence of nepotism, simulating an emergency queen replacement situation, and recording the potential criteria of new queen selections from the larval stage, capping stage, and exit from the chamber stage. The results from this study will aid in the understanding of a colony’s selection of queen candidates in emergency situations.

## 2. Materials and Methods

### 2.1. The Colony without Nepotism

Sixteen colonies were used in this study. Eight were kept at the Honeybee Research Institute of Jiangxi Agricultural University (28.46° N, 115.49° E) as egg-laying colonies. Eight colonies were purchased from Gaoan (Jiangxi, China; 28.14° N, 115.29° E) as queenless colonies. The distance between the two locations is 42 km, which ensures no relatedness between colonies. 

Eight empty combs were chosen (no brood, pollen, or nectar) and divided into four areas with double-sided tape (Figure 1a). A square queen excluder (21 cm × 21 cm) divided one half of a comb into two areas. A queen from each colony was confined to one area to lay eggs for six hours, after which the queen was separated from the experimental hive using a vertical queen excluder. On days 2–4, the queen was controlled to lay eggs in the other three restricted areas using the same method. The larvae in the four regions were thus 1-day-old, 2-day-old, 3-day-old, and 4-day-old on the seventh day after the start of the experiment.

On the sixth day, queenless colonies were prepared. Each queenless colony had one comb with a lot of honey and approximately 10,000 workers. On the seventh day, we added one of the above experimental combs to a queenless colony, respectively.

### 2.2. Number of Capped Queen Cells and Newly Emerged Queens

The number of capped queen cells was counted 13 days after the experimental comb was placed in the queenless colony. Then, the number of queens that emerged from the natural queen cells on the first day was counted. The number of capped queen cells and newly emerged queens were analyzed by one-way ANOVA using SPSS 25.0 (IBM, SPSS, Inc., Chicago, IL, USA).

### 2.3. Nursing Behavior of Worker and Physiological Indicators of Queens

The birth weights of the newly emerged queens were measured with a Precise Electronic Balance (Mettler Toledo, ME204, 0.1 mg). Then, the newly emerged queens were placed in a queen cage and hung in the original colony (>1-day-old). We then counted the number of nurses feeding each newborn queen. Thereafter, each new virgin queen was killed and dissected, and one of its ovaries was used to determine the expression level of reproduction-related genes, while the other ovary was used to make paraffin sections according to the methodology of Yao Yi [12]. The number of ovarioles was counted and analyzed with one-way ANOVA using SPSS 25.0.

### 2.4. Gene Expression Analysis

Total RNA was extracted from the sampled ovaries using the Trans-Zol Up RNA Kit (TransGen Biotech, Beijing, China). The concentration and purity of the RNA were measured using a nucleic acid protein analyzer (IMPLEN). The integrity of the RNA was determined by running an aliquot on a 1% agarose gel. Thereafter, the RNA was used to synthesize cDNA using a reverse transcription kit (TaKaRa, Tokyo, Japan). The housekeeping gene *gapdh* of *A. mellifera* was used as an internal reference gene. Quantitative real-time PCR primers were designed based on mRNA sequences acquired from the NCBI database of the *vg* (vitellogenin LOC406088), *hex110* (hexamerin 110 LOC551648), *jh* (Juvenile hormone, LOC406066), and *gapdh* (glyceraldehyde-3-phosphate dehydrogenase LOC410122) genes in Primer Premier 5.0 (Premier Biosoft). The primer sequences are listed in Table 1.

The qRT-PCR reaction system (10 μL) consisted of 1 μL of cDNA, 5 μL of TB Green, 3 μL of ddH2O, 0.4 μL each of forward and reverse primers, and 0.2 μL Rox. Amplification conditions were as follows: 50 °C for 2 min, pre-denaturation at 95 °C for 10 min, followed by 40 cycles of 95 °C for 10 s, (*vg* at 58 °C; *hex110* at 56.9 °C; *jh* at 58 °C; and *gapdh* at 52.9 °C) for 1 min. For each sample, the specificity of the PCR amplification was verified by melting curve analysis.

Each reaction had four technical replicates for each of the *vg*, *hex110*, and *jh* genes and expression levels were calculated using the 2^−(∆∆Ct)^ method [14]. The expression differences of the *vg*, *hex110*, and *jh* genes were analyzed by one-way ANOVA in SPSS 25.0.

## 3. Results

### 3.1. Number of Capped Queen Cells

The results verify that the number of capped queen cells containing 1-day-old larvae was significantly higher than those containing 3-day-old larvae (Independent-Samples Kruskal–Wallis Test, N = 87, *p* < 0.05). However, there was no difference between the queen cells with 2-day-old larvae and 3-day-old larvae (Independent-Samples Kruskal–Wallis Test, *p* > 0.05) (Figure 2a). Four-day-old larvae do not have capped queen cells. The statistical analysis results are shown in Appendix A.

### 3.2. Number of Newly Emerged Queens

The number of emerging queens reared from 1-day-old larvae was significantly higher than the queens reared from 2-day-old and 3-day-old larvae (Independent-Samples Kruskal–Wallis Test, N = 65, *p* < 0.05). There was no significant quantitative difference between the number of emerging queens reared from 2-day-old or 3-day-old larvae (Independent-Samples Kruskal–Wallis Test, *p* > 0.05) (Figure 2b). The statistical analysis results are shown in Appendix A.

### 3.3. Quantity Statistics of Newly Emerged Queens on the First Day

On the first day, the number of emerging queens reared from 1-day-old larvae was significantly higher than the queens reared from 2-day-old and 3-day-old larvae (Independent-Samples Kruskal–Wallis Test, N = 37, *p* < 0.05). There was no significant quantitative difference between the number of emerging queens from 2-day-old and 3-day-old larvae (Independent-Samples Kruskal–Wallis Test, *p* > 0.05) (Figure 2c).

### 3.4. Queen Birth Weight

There was no significant difference between the birth weights of queens reared from 1-day-old, 2-day-old, and 3-day-old larvae (ANOVA, df_2,34_, *p* > 0.05) (Figure 2d). 

### 3.5. Nursing Behavior and Ovariole Numbers of New Queens 

Data showed that 1-day-old queen larvae triggered more worker followers than 2-day-old queen larvae (ANOVA, df_2,34_, *p* < 0.05), but 2-day-old and 3-day-old queen larvae did not attract significantly different numbers of followers (ANOVA, *p* > 0.05) (Figure 2e).

The number of ovarioles in newly emerged queens from 1-day-old, 2-day-old, and 3-day-old larvae showed no significant difference (Independent-Samples Kruskal–Wallis Test, N = 37, *p* > 0.05) (Figure 2f).

### 3.6. Expression of Genes Related to Ovarian Development

The relative expression of the *vg* gene in queens reared from 1-day-old larvae was significantly higher than that in queens reared from 3-day-old larvae (Independent-Samples Kruskal–Wallis Test, N = 37, *p* < 0.05) (Figure 3a). In addition, the expression of the *hex110* and *jh* genes in queens developed from 1-day-old larvae was significantly higher than those in queens reared from 2-day-old and 3-day-old larvae (Independent-Samples Kruskal–Wallis Test, N = 37, *p* < 0.05) (Figure 3b,c).

## 4. Discussion

Larval age at the time of selection for queen rearing is an important factor affecting queen quality. In this study, in the absence of nepotism, we evaluated queens reared from 1-day-old, 2-day-old, 3-day-old, and 4-day-old larvae. The results confirmed that 3-day-old or younger larvae were selected for queen rearing (Figure 1c), as is widely understood [15,16]. After placing the experimental comb in the queenless colony, we found that the queen cells began to appear in the 1-day-old larvae area, suggesting that 1-day-old larvae are considered queen candidates. Sagili et al. [17] showed that honey bees perceive the nutritional state of larvae and use that information when selecting larvae for rearing queens in the natural emergency queen replacement process. In addition, Margarita and Rangel et al. reported that workers recognize larval hormones to distinguish between the larval grades and developmental stages [18,19], suggesting that both nutrition and hormones may be important in screening queen candidates in the larval stage. Unfortunately, the behavioral aspect of the workers’ selection of queen candidate larvae was not captured. However, recent reports have suggested that learning behavior in bees starts at an early age, and whether it includes judgment is worth considering [20].

It is known that not all the queens from capped queen cells will emerge, as some are destroyed by workers before or after capping [14,21,22,23]. In this study, the worker’s removal of capped and uncapped queen cells started on the sixth day and did not target a particular larval age (Figure 1b,c). This suggests that workers consistently select queen candidates based on quality criteria. Here, the workers cleared the uncapped queen cells because the larvae had stagnated or died, and the same might be true for the capped queen cells (Figure 1d). Morse et al. [24] also found that if the larvae were removed from a capped queen cell, the workers would remove the queen cell. We found that the number of capped queen cells containing 1-day-old larvae was significantly higher than those containing 3-day-old larvae (Figure 2a). Data suggest that workers prefer 1-day-old larvae for filling queen cells, probably because 1-day-old larvae develop into higher quality queens. Rehman et al. [25] used different-aged larvae (12–24, 24–48, and 48–72 h) for grafting, and found that 1-day-old larvae had the highest acceptance rate. Thus, the selection of queen candidates by the colony adheres to the initial selection criteria during the larval stage, but is also a gradual process of optimization.

Interestingly, queens reared from different-aged larvae (1-day-old, 2-day-old, and 3-day-old) all emerged on the first day (Figure 2c), but most of these were from 1-day-old larvae. This may be because queen cells first appeared in the 1-day-old larvae area and these larvae first consumed royal jelly, promoting larval development. Ucak Koc [26] found that the 10-HDA content of royal jelly decreased with increasing larval age in different day-old larvae transferred to the queen table, which supports our findings. This indicates that when colonies screen queen candidates from larvae of different ages, they prioritize the 1-day-old larvae as good quality. However, because of the particularity of the queenless colony, they also select good-quality older larvae (3-day-old larvae) as candidate queens. 

During the emerging stage, we placed newly emerged queens into the original colony and allowed nurse bees to feed them. We found that the queens that developed from 1-day-old larvae had more nurse bees. Many studies also found that 1-day-old queen larvae triggered more worker followers than 2-day-old and 3-day-old queen larvae [19,27,28,29]. Thus, an older larval queen is an emergency measure, and the colony will choose the most suitable queen for the growth of the colony.

Various studies have shown that the reproductive potential of queens can vary quite considerably and can be measured in their size, number of ovaries, and weight [16,30,31,32,33,34,35]. The expression of the *vg*, *hex110*, and *Jh* genes can also be used to determine the quality of queens, as they are related to physiological indicators and ovarian activation [36,37,38]. Interestingly, there was no significant difference in the birth weights and number of ovarioles of queens reared from 1-day-old, 2-day-old, and 3-day-old larvae (Figure 2d,f), but our results showed that the relative expression levels of the *vg*, *hex110*, and *Jh* genes were higher in queens reared from 1-day-old larvae than those reared from 3-day-old larvae. This suggests that the quality of the queens reared from 1-day-old larvae was superior (Figure 3). 

## 5. Conclusions

While the exact mechanism by which workers assess queen quality is unknown, the result of this layered filtering of investments in queen candidates may be to maximize colony development. In the absence of nepotism, the selection of queen candidates by workers is not final but is a gradually optimized process. Thus, workers can play an important role in the quality control of the queen selection process, which is highlighted by the results of this study.

## Figures and Tables

**Figure 1 animals-14-01535-f001:**
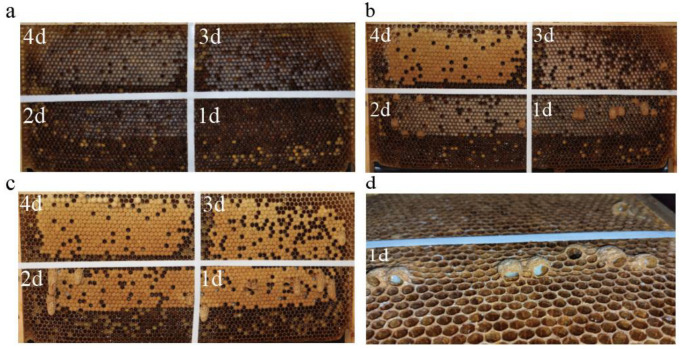
Honey bee comb full of larvae. (**a**) Comb full of larvae on the seventh day after the start of the experiment; (**b**) uncapped queen cells; (**c**) capped queen cells; (**d**) larval state.

**Figure 2 animals-14-01535-f002:**
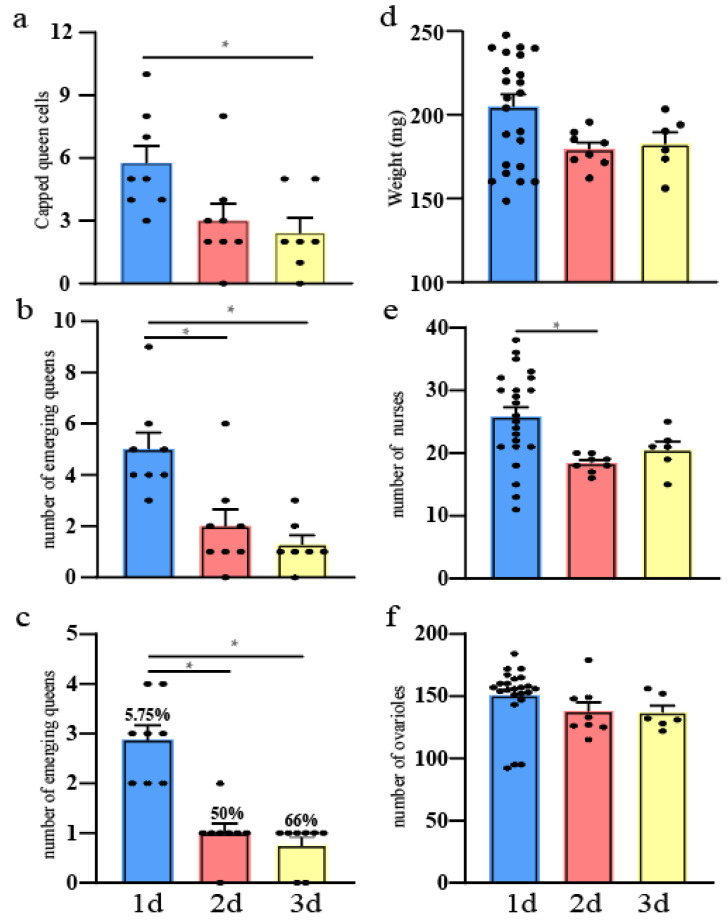
Analysis results of honey bee queens reared from different-aged larvae in a non-nepotistic colony under emergency conditions. (**a**) The number of capped queen cells; (**b**) the number of newly emerged queens; (**c**) the number of newly emerged queens on the first day, percentage: newly emerged queens on the first day/newly emerged queens; (**d**) the birth weight of queens; (**e**) the number of nurses in the original colony; (**f**) the number of ovarioles in queens reared from different-aged larvae; the horizontal axis represents the emerging queens from the different ages of larvae; * indicates significant differences (*p* < 0.05).

**Figure 3 animals-14-01535-f003:**
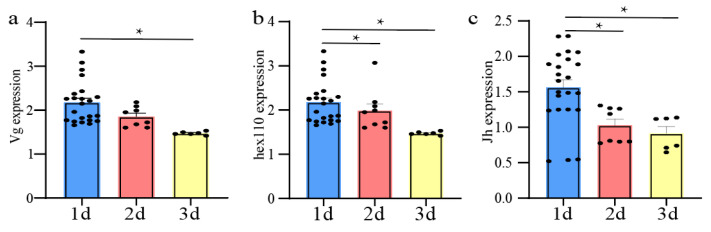
Expression of the *vg*, *hex110,* and *jh* genes in the honey bee queens reared from 1-day-old, 2-day-old, and 3-day-old old larvae in a non-nepotistic colony under emergency conditions. (**a**) Expression of the *vg*; (**b**) Expression of the *hex110*; (**c**) Expression of the *jh*; * indicates significant differences (*p* < 0.05).

**Table 1 animals-14-01535-t001:** Primers designed for the qPCR analysis of the expression levels of honey bee genes related to ovarian development, and their sequences.

Target Gene	Forward Primer (5’-3’)	Reverse Primer (5’-3’)
*gapdh*	GCTGGTTTCATCGATGGTTT	ACGATTTCGACCACCGTAAC
*hex110*	AACGTGCCAGGCGCAGTTGT	TTCACCAGCATGGAGGTTCTGGA
*jh*	CACTGGCACCAGAGCCTGTC	GATTCCCATTGAACGAGCGA
*vg*	CGTGTTCCAGAGGACGTTGA	ACGCTCCTCAGGCTCAACTC

## Data Availability

The data presented in this study are available in this article.

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
