# Peer review of "The Honey Bee Colony’s Criterion for Candidate Selection: “Ongoing” or “One-Shot”?"

_animals, 2024, doi:10.3390/ani14111535_

Round 1
Reviewer 1 Report
Comments and Suggestions for Authors
The paper entitled “The honey bee's colony's criterion for candidate selection: ‘ongo-2 ing’ or ‘one-shot’? “ provides information on queen candidates in colony growth. The authors conducted a requisite research for this topic. The results are clear and respond to the objectives of the study. Therefore, I recommend their publication in this journal after considering the following suggestions:
In the value of the significant p < 0.05, the authors have written different ways of writing the letter P. Could the authors unify the letter p in lowercase and italics?
In line 206, it is necessary to add the word "honey" in front of "bees," as this is a honey bee colony.
From line 223, the authors use a new way to express "1-day-old," which is written as "1d." It is unclear whether this is due to a reference that uses this way of expressing the number of days.
Reviewer 2 Report
Comments and Suggestions for Authors
The paper entitled “The honey bee colony's criterion for candidate selection: "ongoing" or "one-shot"?” by Pan et al. deals with the selection of queen candidates by workers as a gradually optimized process, in the absence of nepotism. Interestingly, the 1-day-old larvae are preferred as queen-candidates. The queens emerged from such larvae attract a higher number of nurse bees and, despite their ovaries have a similar number of ovarioles and birth weights with respect those of queens reared from 2-day-old, and 3-day-old larvae, they show a higher level of espression of reproductive related genes such as vg, hex110, and Jh genes. The experiments are well designed. All the sections of the MS are clearly described or discussed. Only some minor corrections have to be done, listed below. The paper merits to be published in Animals.
· In the “Results” section it has not mentioned that the 4-day-old larvae were never selected for queen rearing. Although Authors reported this information in the Discussion, it should be added in the previous section. In the Discussion, this statement may confirm the literature data.
Line 143: the indication of (Fig. 2-a) should be inserted before the phrase “The statistical analysis results are shown in Supplementary Table S1”
· Please correct as……Wallis Test, p > 0.05) (Fig. 2-a). The statistical analysis results are shown in Supplementary Table S1.
The same observation is also valid for the indication of the Fig. 2-b at the line 151.
· Please correct as……. (Independent-Samples Kruskal-Wallis Test, p > 0.05) (Fig. 2-b). The statistical analysis results are shown in Supplementary Table S1.
Fig. 2 b, c, e, f:
· please, delete the determinative article “the” in the ordinate axis.
In the caption of the Fig. 2, lines 177-178, I was unable to find “The different lowercase letters”.
· Please, in analogy with the caption of the Fig. 3, correct as …*indicate significant differences (P < 0.05).
Line 201:…Hatch et al.
· Please, indicate the reference number [16].
Line 219: Nabeel Ur Rehman et al [25]
· Please, delete Nabel Ur. Change as Rehman et al [25]
Lines 234-235: Long et al. also found that 1-day-old queen larvae triggered worker followers than 2-day-old and 3-day-old queen larvae [19,27-29].
· Why only Long et al. (eventually, add the reference number 28) have been cited for the adfirmation that “1-day-old queen larvae triggered worker followers than 2-day-old and 3-day-old queen larvae”, if other three references have been indicated (19, 27 and 29)? Please correct the phrase accordingly. Please, also insert “more” between the words triggered and followers. “1-day-old queen larvae triggered more worker followers than 2-day-old and 3-day-old queen larvae”….
